# "Bringing greater research fluency into our educational vision": A qualitative research study on improving Traditional Chinese Medicine research education

Heidi Most [1]*, Lisa Conboy[2,3]☯, Rosaleen Ostrick[4,5]☯, Belinda J. Anderson[6,7]☯

1 Department of Acupuncture and Herbal Medicine, Maryland University of Integrative Health, Laurel, Maryland, United States of America, 2 Department of Gastroenterology, Beth Israel Deaconess Medical Center, Harvard Medical School, Boston, Massachusetts, United States of America, 3 Research Department, Seattle Institute of East Asian Medicine, Seattle, Washington, United States of America, 4 Jules Stein Eye Institute Retina Division, University of California Los Angeles, Los Angeles, California, United States of America, 5 Western Medicine Division, Yo San University, Los Angeles, California, United States of America, 6 College of Health Professions, Pace University, New York, New York, United States of America, 7 Department of Family and Social Medicine, Albert Einstein College of Medicine, New York, New York, United States of America

☯ These authors contributed equally to this work.
* mostheidi1@gmail.com

**Data Availability Statement:** All underlying data are available from the Figshare database (https://doi.org/10.6084/m9.figshare.27056236.v1).

## Abstract

The main objective of this qualitative analysis is to increase the quality of Traditional Chinese Medicine (TCM) research education. Subject matter experts and stakeholders were sent a list of possible topics to include in a research course and were asked to comment on it, add or subtract topics, and suggest appropriate research papers to use, lessons learned in their topic areas and possible stand-alone topics for inclusion in other courses. Their feedback was used to revise the topic list, create a list of foundational research papers to include in a research course and devise a list of stand-alone topics that could be included in the general TCM research curriculum. The result is a TCM research course curriculum that will improve the teaching of evidence-based medicine (EBM) practices to TCM students. This will help TCM faculty and program directors improve existing courses, develop new courses and strengthen EBM education.

## Introduction

TCM, including acupuncture and Chinese herbal medicine, is taught and practiced worldwide in approximately 196 countries and regions [1]. As of mid-2024, there are 49 acupuncture schools in the United States (U.S.), offering master's and clinical doctoral degrees in acupuncture and Chinese herbal medicine [2]. In 2023, there were approximately 34,000 licensed U.S. acupuncturists [3]. Although the U.S. acupuncture profession is considering renaming the profession, the term used in this paper is TCM because it is recognized internationally.

**Funding:** The author(s) received no specific funding for this work.

**Competing interests:** The authors have declared that no competing interests exist.

The aim of this research was to improve the teaching of acupuncture research in U.S. acupuncture schools. Evidence-based medicine (EBM), including knowledge of current best clinical practices, clinical experience and patient preferences is critical to biomedical and complementary healthcare [3]. Teaching acupuncture students to understand and use these three areas is critically important in their development as safe and effective practitioners [4]. There has been an exponential increase in acupuncture research over the past 20 years [5]. A total of 13,320 acupuncture-related publications were identified in that time, with an annual growth rate of 10.7%, compared with 4.5% in biomedicine [5]. This research includes basic science studies investigating mechanisms and clinical research demonstrating the efficacy and effectiveness of acupuncture for a wide range of conditions [6].

To increase the quality of research education in Complementary and Integrative Health (CIH) institutions, nine collaborative institutional partnerships were supported by the National Institutes of Health's (NIH) National Center for Complementary and Integrative Health (NCCIH) in 2005–2011. A primary goal was to increase CIH students' and faculty's exposure to and understanding of research literature to improve research literacy among CIH practitioners [4, 7]. This support produced a wealth of resources, including methods for faculty training and development, EBM and research literacy competencies, different instructional resources for faculty and students, and identified implementation challenges [8]. In addition, the Academic Collaborative for Integrative Health (ACIH—now part of the Academy of Integrative Health and Medicine) developed a Project to Enhance Research Literacy (PERL)—producing a web-based repository of resources for developing evidence-informed research curriculum [9]. Their resources include curricula for students and faculty and examples of syllabi.

The Accreditation Commission for Acupuncture and Herbal Medicine (ACAHM) has developed standards for research and evidence-informed practice education [2]. The entry-level (master's degree) competencies center around the ability to find, assess and use biomedical research as part of a treatment plan. Following an open public comment period, the master's level research competencies were expanded to develop research competencies for the new doctoral-level programs that started in 2013. These competencies included the ability to "discuss East Asian Medicine in terms of relevant scientific theories" and "assess relevant information from a wide variety of sources to support the education of colleagues" [2].

Research on the effectiveness of research education in East Asian medicine (TCM) degree programs in the United States (US) [8, 10–12] has identified unique challenges associated with educating TCM students and faculty about research. These are related to the somewhat negative perspective many TCM students and faculty have toward biomedical research on acupuncture [4, 13–15]. Some of this stems from the use of the randomized controlled trial (RCT) and associated methodological concerns that render such research less reflective of real-world clinical practice [16, 17]. Other issues relate to the devaluing of thousands of years of well-documented theory that informs clinical practice and the tendency of biomedicine to co-opt traditional medical practices [18]. These factors partially explain the minimal extent to which many US TCM schools have incorporated robust research curricula into their degree programs. Other more pragmatic factors include the fact that many TCM schools do not have the financial resources to support comprehensive curriculum improvement or faculty with appropriate research education to develop and implement research curricula. Before the development of doctoral degree programs in 2013, the limited research curriculum accreditation requirements for master's degree programs often rendered such programs almost devoid of research content. This lack of research content was a significant factor in the NIH developing the R25 grant mechanism to support improvement in research and EBM content in these degree programs [7].

Despite these challenges, acupuncture research and practice have many strengths. As was mentioned previously, there is an abundance of acupuncture research [5]. Acupuncture research has made important and innovative contributions to medical research at large, not just complementary medicine [19]. The considerable challenges in researching acupuncture, such as the design of placebo (often referred to as sham) controls, have been well documented [20–22]. Many stakeholders in the US are committed to expanding and improving the profession, including national associations, TCM educational institutions, and individual educators and clinicians.

In line with its mission of advancing and disseminating scientific research into acupuncture and TCM to inform global health care, the Society for Acupuncture Research (SAR) facilitated the creation of Special Interest Groups (SIGs). The SAR SIG-Edu aims to improve the teaching of research education in TCM degree programs by creating a list of research topics to include and providing TCM schools with accessible and useful resources to develop research curricula. It is hoped that such improvements in research and EBM education will enhance the participation of TCM experts in research projects, legislative initiatives, and clinical implementation of TCM within mainstream healthcare. This paper describes a qualitative study of subject matter experts (SMEs) and stakeholders to determine topics that should be included in a research course for TCM degree programs and best approaches and considerations in educating TCM students about research.

## Methods

### Survey Instrument

The development of the model curriculum began with the formation of the SAR SIG-Edu, continued with creation of a proposed list of topics to be included in a model acupuncture curriculum, and resulted in a survey requesting feedback from SMEs and stakeholders. See S4 Survey Instrument Details for more information.

Different questions were asked of SMEs and stakeholders because of their different roles. SMEs are primarily researchers interested in specific specialties (e.g., cancer care, pain), whereas stakeholders are primarily educators interested in teaching and in the TCM profession as a whole. Three of our respondents were both SMEs and stakeholders and were counted in both categories. SMEs were asked to comment on the model curriculum, suggest three papers in their areas of expertise, and summarize three to four lessons learned in their area of expertise. The stakeholders were asked to comment on the model curriculum, suggest other topics for inclusion, and identify topics that could be stand-alone modular presentations. Table 1 lists the requests made to the SMEs and stakeholders. SMEs were assigned codes R1-R25, and SMEs were assigned codes R26-R34. The three respondents who were both SMEs and stakeholders were given two different numbers to differentiate their contributions as

**Table 1. Survey instrument development timeline.**

| Months | 1/2019-6/2019 | 9/2019-12/2019 | 11/2021-2/2022 | 3/2022-9/2022 |
|---|---|---|---|---|
| SAR SIG-Edu formed at SAR Conference | X | | | |
| Members from US and Brazil contributed to major topics to a Google doc to be included in an acupuncture research curriculum | | X | | |
| Identified SMEs and Stakeholders and requested feedback on model curriculum | | | X | |
| Feedback received and analyzed | | | | X |

SMEs and stakeholders (R10/R29; R17/R33; R25/R29). See S4 File for the different questions asked to SMEs and stakeholders.

## Survey implementation

The SAR Board reviewed our process and agreed to co-sign a letter with the research team that was emailed to the SMEs and stakeholders to solicit their feedback. The letter included the Model Curriculum developed by the SAR SIG-Edu and the specific questions outlined in S4 File.

## IRB/informed consent

This project was approved by the Maryland University of Integrative Health institutional review board (study ID: 01.MOS.02.21.1). Response to our email request indicated informed consent.

## Author credentials and experience

Lisa Conboy, MA, MS, ScD, has been studying Chinese Medicine and teaching Chinese Medicine students research methodology at schools across the US for over 25 years. Heidi Most, DAc, LAc, is a Doctor of Acupuncture and a licensed acupuncturist and professor who has taught acupuncture theory, research, and clinical courses at the Maryland University of Integrative Health for 22 years. Rosaleen Ostrick, MPH, MA, MATCM, is Administrative Director, Department of Ophthalmology-Retina Division, UCLA, where she is responsible for clinical research. She is also on the faculty at Yo San University and a licensed acupuncturist. Belinda (Beau) Anderson, PhD, MA(Ed.), LAc, started her career as a molecular biologist and later studied TCM. She is an NIH-supported researcher, licensed acupuncturist in part-time practice, and Associate Dean and Professor in the College of Health Professions at Pace University, New York City. From 2006 to 2018, she was Academic Dean at Pacific College of Oriental Medicine (now Pacific College of Health and Science) and is still on the faculty.

## Data analysis

Results were collected by HM in a single document that was emailed to the other three primary research team members (BA, RO, LC). All four team members were tasked with independently developing a list of major themes expressed in the participant responses. Each member presented their list in a meeting in July 2022. Discussion led to consensus on a consolidated list of major themes. The authors wrote their report using the Standards for Reporting Qualitative Reports (SRQR) [23].

In addition to determining major themes, HM used the responses from nine SMEs and stakeholders to reorder the topics of the original curriculum. The revised curriculum was then reviewed and agreed to by the other authors.

## Word cloud

Word clouds are used in qualitative research to represent participant feedback visually [24, 25]. HM used a free Word Cloud generator [26]. A list of single words was created to represent each SME and stakeholder's comments on the overall curriculum. Some phrases were combined into single words in order to remain together. These words were entered into the word cloud generator on 7/8/23, and the resulting word cloud was copied into this article.

**Table 2. Responses from the stakeholder organizations.**

| Stakeholders that were asked to comment | # who responded |
|---|---|
| NCCAOM | Its accrediting board prohibits participation in educational requirements for the acupuncture profession |
| CCAHM: 4 people asked | 1 responded |
| ACAHM: 2 people asked | 1 responded |
| SAR: 1 person asked | 1 responded |
| ASA: 3 people asked | 0 responded |
| Medical Acupuncturists: 1 person asked | 1 responded |
| SAR SIG-Edu: 24 people asked | 2 responded |
| Individual researchers who are also stakeholders from past roles as educators: 3 people asked | 3 responded |

## Results

Multiple people in six stakeholder organizations were sent the emails. Responses were received from none, one, or two people from each organization. A group email also queried the entire SAR SIG-Edu email list of 24 people. Two people from that group responded. Three respondents were both SMEs and stakeholders. They were counted twice in the totals as they answered both sets of questions. There was a wide range of variability in the length of responses to each question. One person said this was due to a lack of time. Others gave no reason. Data is presented in Table 2.

Replies were received from 17 of the 25 SMEs (68%). Replies were received from 9 out of 38 Stakeholders (24%). Data is presented in Table 3.

### Respondent's views on the overall curriculum

Twenty-three people answered this question including 16 SMEs and 7 stakeholders. Twenty-one told us that the proposed curriculum was very good. Some comments were: R1 - "Amazing," R9 - "Really excellent. . .important," R23 - "covers the needed topic areas," R10/R29 - "selection of topics is excellent and they all seem necessary," R21 - "good list of topics," R12 - "such a standardized program will provide students an appreciation of EB practices in integrative medicine," R8 - "curriculum looks really good–should be widely spread," R31 - "Impressive work. Thank you so very much for bringing greater research fluency into our educational vision, "R25/R28 - "Wow, huge thanks for finally doing this much-needed work!. . . Choice of topics is spot on."

Two people (one SME and one stakeholder) had problems with the proposed curriculum.

R6 - ". . .this curriculum is about 'teaching the scientific facts on acupuncture', and that often has the feeling of reassuring students that acupuncture has the 'official seal of science'.

**Table 3. Summary of responses.**

| | SMEs | Stakeholders | Total respondents |
|---|---|---|---|
| | n | n | n (%) |
| Asked | 25 | 38 | 63 (100%) |
| Responded | 17 | 9 | 26 (41%) |
| Consented to be named | 11 | 6 | 17 (27%) |
| Answered first question regarding their opinion of overall curriculum | 16 | 7 | 23 (88% of responders) |
| Positive to curriculum | 16 | 6 | 21 (91% of those who answered first question) |
| Negative to curriculum | 1 | 1 | 2 (8% of those who answered first question) |

It is not actually focusing on the key point about research: finding out something about the world, especially if that something might be contrary to your prior beliefs."

R27 - "Your list does not follow any convention I recognize for a curriculum outline. In general, I find it oddly specific and then very general."

A word cloud representing the combined responses is presented in Fig 1.

## Major themes identified in respondents' views on the overall curriculum

**1. Worldview/epistemology.** TCM has a different worldview from Biomedicine (BM). One of the strongest themes expressed by our respondents is that students need to understand that there are different ways of knowing and that each is valid and effective in its own realm.

R 10 - "Understanding how research using the scientific method is ONE way of knowing an epistemology. . .Chinese medicine provides a different way of knowing. To use the words of biology and First World plant women Robin Wall Kimmerer. . . We need to make sure our students . . . "have an awareness of other ways of knowing, they have this glimpse into a worldview which is really different from the scientific worldview. I think of them as just

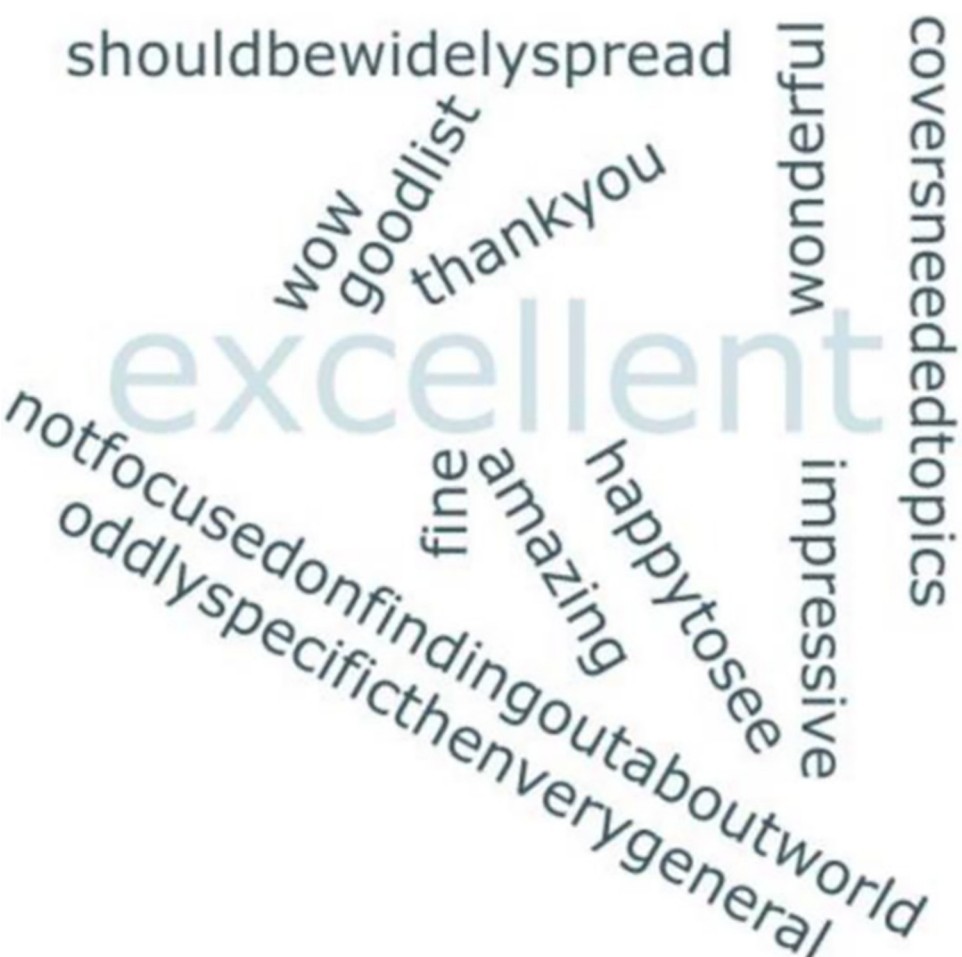

**Fig 1. Word cloud of participant responses describing the model curriculum.**

being stronger and having this ability for what has been called "two-eyed seeing," seeing the world through both of these lenses, and in that way, have a bigger toolset for environmental (medicine) problem-solving."

Four respondents felt strongly that acupuncture research education needs to start by informing students of different worldviews and how biomedical research is just one worldview and way of looking at TCM.

R13 - "[TCM students'] clinical training emphasizes holistic approaches to diagnosis and treatment. As such, clinical research often appears extremely one-sided and overly simplistic. For this reason, the context and epistemology of science and the role of evidence-based medicine must be presented to students as a kind of disclaimer before exposing them to the details of the methodology and outcomes of clinical research as it has been applied to East Asian medicine."

R17/R33- "CM was [historically] experience based. Started with people who were incredible observers; kept what worked. Also, the idea of being more open to experiences. Ways of knowing include classical and modern texts. Don't want an MD who only has books from the 1950s. Clinical experience is a way of knowing. Research is another way of knowing; intuition is a way of knowing."

R16/R34 - "Start with Different Ways of Knowing module. Understand research is one way of knowing. CM was experienced based. Started with people who were incredible observers; kept what worked. Also the idea of being more open to experiences."

**2. Challenges with researching acupuncture.**   R21 - "[t]he multifaceted intervention that is acupuncture. . . changes over time. As clinicians we know what we do–delivering a tailored intervention that meets [patients'] needs and changes over time."

Another respondent suggested that complexity science may provide a method for researching TCM treatment. R21- "Complexity science is not being used in health science. So we don't really know how to use it for research–methods are not up to speed in the health sciences. . .they are used in other sciences."

The importance of including qualitative research was mentioned—R17/R33 - "Explore the medical model of acupuncture; talk about data on what patients want and receive and why, and what acupuncture practitioners intend, and like about their work."

A respondent added this comment regarding placebo (sham) acupuncture–R19 - "I think it is important to note that the question of placebo effects has never been an issue in traditional East Asian Medicine. In my opinion, it undermines fundamental epistemological premises on EAM."

**3. Challenges with teaching acupuncture research to TCM students.**   Seven of our respondents commented on these challenges. Some examples, grouped by similarity, include:
R17/R33 - "[Students] do not arrive at acupuncture schools eager to learn about clinical trials and lab work on mice. Their goal is to serve patients—human beings—not to gather scientific data, or they would likely be studying elsewhere."
R23 - "Most students have little research background." R27 - "My impression is that most acupuncture students at the majority of schools do not have sufficient physiologic training to

understand your physiology topics." R25 - ". . . 'basic [research] literacy' is a big expectation for Masters students, particularly those at the smaller schools".

R23 - "How to incorporate the knowledge of EBM on acupuncture into other courses is a challenge because most instructors of these courses have not been trained in EBM and acupuncture clinical research." In addition, this respondent added that due to the level and understanding of research by many students ". . .discussion about the challenges, particularly about "negative" trials should be very cautious. More effort should be put on the "positive" trials emphasizing the discovery of acupuncture research."

## Lessons learned in SME's area of expertise

SMEs were asked to summarize three or four lessons in their area of expertise. Some of the responses are supportive of the themes found above. Fifty-six percent of SMEs responded to this question. Examples of their responses, grouped according to similarity, are shown below.

R9 - "Current evidence of the effectiveness of acupuncture."

R7 - "High quality pragmatic trials can be done in usual care settings and results can inform stakeholders such as statutory health insurance companies on an international level."

R12 - "We need to better understand how acupuncture treatment integrates with conventional medical treatments."

R7 - "The context matters (e.g. patients' treatment outcome expectation, patient-practitioner interaction)." R2: "[T]here's much we don't know about acupuncture and patient/ acupuncturist therapeutic relationships that can impact efficacy." R12: "An individual patient's physiology matters, as effectiveness may be impacted by age, gender and physiological condition." R19: "Evidence suggests acupuncture's placebo effects is larger than other types of placebos. The patient-clinician relationship is a key component of acupuncture and sham acupuncture treatment." R16/R34: "Another understanding is that we are all connected. Universe is a cosmic koan. 'The universe is finite but with no boundary'–Stephen Hawking."

R7 - "In chronic pain diseases sham controlled trials need very large samples because group differences are small."

R2 - "[T]he brain is not a static or linearly degenerating organ (in adulthood). It is plastic. Chronic pain can change the brain, but relief from pain can also change it in important and profound ways."

R16/R34 - "Still don't know how intention affects physiology. Are we all connected? What are we connecting with? How does this impact our acupuncture treatments?. . . The important distinction is to talk about correlates of acupuncture. . .we have no idea how to turn meaning into physiology. Placebo includes expectation. . . .[E]ven with open placebo, they still benefit."

## Papers SMEs would want students to read in their area of expertise

S1 File lists all the recommended papers that SMEs felt were critically important in their field. It was noted that because acupuncture research is evolving, any list of research papers would need to be regularly updated.

### Topics stakeholders suggested as modular stand-alone presentations that could be inserted into existing courses

Whole Person Research/Whole Systems Research (R21, R25/R28, R31)

Complexity Research (R17/R33, R21, R25/R28, R31)

Health Disparities (a whole course as well as throughout the curriculum) (R31)

Designing a research study as an exercise in applied knowledge, taking the student step by step from a clinical observation or theoretical statement to the steps in the design of a research question, hypothesis, and design of a pilot study (R10/R29)

### Topics stakeholders suggested could be added to the proposed curriculum

Many important and diverse suggestions were received from the respondents. These proposed topics were added under section VII of the revised model curriculum shown below.

### Final revised model curriculum

Taking SME and stakeholder comments into account, our original list of topics to be included in an advanced acupuncture research course (the model curriculum) was reordered and revised, as shown in Table 4. Most of the responses explicitly stated or implied the importance of research as a part of EBM, so this topic was added and placed as the first topic. Based on the theme analysis outlined above, Worldviews was added as the second topic. Because several respondents talked about the importance of making the research relevant to students' lives, we moved the topics of Racial Disparities in Health and Health Care, and Whole Person Research earlier in the curriculum and moved the Physiology of Acupuncture to later. Additional proposed topics were given their own section (VII) and can be selected by individual degree programs or faculty to be included in specific degree program courses. The original curriculum is presented in S2 File.

## Discussion

Our survey yielded a wealth of information, including major lessons learned in SME's topic areas and recommendations for foundational journal papers to use in teaching. The survey results also validated the SAR SIG-Edu draft model curriculum.

The survey uncovered a dichotomy in TCM as practiced vs TCM as researched. Most of the SMEs are researchers, and their interest is typically collecting quantifiable scientific data. Most of the stakeholders, on the other hand, are primarily clinicians and educators. Their interest is typically focused on how to help students as future clinicians improve their practice through the use of research. The curriculum outlined above seeks to bridge that gap by emphasizing several key issues: 1) differing worldviews can give rise to and support different types of research techniques and subsequent data, 2) an understanding of the data uncovered by bench science and RCTs can inform clinical practice and influence legislative and insurance guidelines; and 3) there are some forms of research (e.g., qualitative and complexity science) that may be better than others in reflecting the real world practice of EAM.

One challenge facing TCM schools is that TCM degree programs typically limit the number of credits that can be devoted to research curricula. Additionally, TCM colleges are challenged in their capacity to develop research curricula due to financial constraints and a lack of faculty with research expertise. Developing a new research course is therefore often unfeasible. However, incorporating these topics into existing courses, especially through the provision of stand-alone modules, might be feasible [4]. Ideally, introductory research topics would be available to TCM students early in their degree program. The SAR SIG-Edu has a repository of course syllabi from its members containing such topics that could be used for this purpose.

**Table 4. Final model curriculum.**

| Topic | Sub-topics* |
|---|---|
| I. Importance of Research and Evidence-Based Medicine | • Intro to EBM<br>• Intro to basic research literacy skills |
| II. Worldview: different ways of knowing | |
| III. Challenges in Acupuncture Research | • Placebo-what it is, understanding placebo effect/other possible controls, e.g. waitlist.<br>• Context effects: acupuncture is an umbrella term with multiple components.<br>• How to research a multi-modal treatment that changes over time in response to a changing patient condition: Complexity Science and Qualitative Research |
| IV. Racial Disparities in Health and Health Care | |
| V. Clinical application of acupuncture | • Whole Person Research: what we have learned from clinical trials on acupuncture's effect on the sense of well-being of patients; qualitative research on patients' perspectives<br>• Chronic and Acute pain<br>• Dental pain<br>• Headache<br>• Back-related pain<br>• Arthritis<br>• Cancer-related pain<br>• Anxiety and depression<br>• Hypertension<br>• Patients' quality of life<br>• The NADA protocol, Battlefield Acupuncture, and community acupuncture<br>• Women's and men's Health |
| VI Physiology of Acupuncture | • Endocrine system<br>• Nervous system/fMRI studies<br>• Immune system<br>• Propagation of signals through tissue<br>• Connective tissue |
| Additional topics suggested by responders | • Basic research literacy<br>• Case report writing<br>• The importance of research on the field of acupuncture, including shaping insurance policy and practice standards<br>• Covid 19 treatment<br>• Emerging technologies (apps, techniques)<br>• Applied research: designing and implementing a research project; incorporating research into clinical practice<br>• Biofield therapies<br>• Complexity Science |

*See S1 File for the papers that SMEs recommended in many of these sub-topics

This could be a prelude to more advanced research courses covering topics in our proposed model curriculum outlined above. Such courses would be suitable for both master's and doctoral degree programs. Several respondents in our study emphasized the importance of engaging students in experiential learning, for example, by having them design a research study. This would be especially valuable for doctoral programs and as preparation for capstone projects. Including research questions on the NCCAOM exam would be a strong incentive for schools to offer these topics and for students to study them. For this to happen the NCCAOM job analysis (that determines current TCM practices of an entry-level practitioner) [27] would need to inquire about the use of research in clinical practice such that questions could be included on the certification exams.

The journal articles recommended by SMEs are foundational publications in their area of expertise. This list is an invaluable resource for both TCM faculty and students in research courses. We hope the list will provide relatively easy access to pertinent, high-quality, scientific literature. Such lists of foundational publications should be updated on a regular basis.

One SME suggested statistics as an additional topic. However, we chose to leave it out of our proposed model curriculum. Basic statistical knowledge is best presented in an entry-level research course. Our proposed model curriculum is focused on more advanced TCM-specific topics, and hence, we did not include basic statistics.

Our perspective is that all students should be taught to engage in reflective thinking with every patient encounter and to periodically capture that thinking in the creation of case reports. We also recommend that students are required to cite research to support treatment strategies and protocols.

Two respondents said this was a list of topics more than a curriculum. We agree that this is a list of topics and that this is one way to define a curriculum. By design, ACAHM standards for master's and doctoral degrees allow programs the flexibility to create courses as they see fit. This list of topics gives flexibility to TCM programs and provides specific guidance about content. Even those schools with a robust research component could benefit from these suggestions.

The response rate from stakeholders (24%) is indicative of the challenges in conducting a national survey. One national organization could not respond due to prohibition by the organization's accrediting body, and one did not respond and gave no explanation. In instances of one response from an organization when multiple people were asked, one person may have been appointed as the spokesperson. Most members of the SAR SIG-Edu did not respond. Possible reasons for non-response include not seeing the email or reminder, not having time, lack of interest, or perceived lack of expertise in the topic.

Simultaneous to our work chronicled above, we also developed and administered a survey of acupuncture students to assess their background and interest in acupuncture research. We based our survey on one done at Pacific College of Health and Science (previously known as Pacific College of Oriental Medicine) in 2012 [15]. The rationale for performing this study was to identify barriers to teaching acupuncture research and opportunities to engage and excite students to study acupuncture research. Understanding students' barriers to learning relevant research is the first step in reducing those barriers by informing strategies aimed at increasing student interest [28]. The full results of this student survey will be published in a separate paper.

Finally, left unanswered is how this valuable information is best disseminated. Dissemination is a critical next step for research on improving TCM research education to be best utilized. Our dissemination plan is to send our model curriculum and findings to all US TCM schools. Dissemination is an important issue in all medical fields—how to ensure that new research discoveries make their way into clinical practice and that clinical practice reflects best practices [29]. By improving research education for acupuncturists, we hope future acupuncture practitioners will more effectively apply research to clinical practice and possibly participate in research, legislative, and insurance initiatives.

## Limitations

This research had several limitations. The authors' varied backgrounds as researchers, acupuncturists and Chinese herbal practitioners influenced their choice of SMEs and their interpretation of the responses. Not all SMEs or stakeholders responded, and not all questions were answered by all respondents. The questions were open-ended, and the answers were diverse,

making consolidating representative themes difficult. One of our authors (BA) was a respondent and also involved in the theme analysis. There was no uniform way to gather data about the research needs of all TCM programs, so we do not know if our data is representative of schools nationally. We also do not know if our findings are applicable or of interest to the TCM educational community.

## Conclusions

The specific contributions this study adds to the existing literature are: 1) a list of topics that TCM programs can use to design a research course, improve an existing research course, or incorporate into existing courses; 2) introduction of a topic (worldview) that will likely improve student engagement with research; 3) a list of foundational research papers which will improve students' understanding of acupuncture mechanisms and physiology; 4) a list of topics that can be developed into stand-alone electronic presentations that schools can plug into existing curriculum. We hope this paper will be useful to TCM program faculty who teach research, and to students in their future professional activities as they become clinicians or pursue a research career.

## Supporting information

**S1 File. Recommended papers from SMEs.**
(DOCX)

**S2 File. Original curriculum.**
(DOCX)

**S3 File. Characteristics of subject matter experts and stakeholders.**
(DOCX)

**S4 File. Survey instrument details.**
(DOCX)

## Acknowledgments

The authors are grateful to SAR for their work in strengthening the acupuncture profession, including providing researchers with an international forum to share and disseminate their work, creating the Special Interest Groups (of which the Education group is one of several) to work on specific areas of acupuncture research, and for providing educational opportunities. The authors are also grateful to all the SMEs and stakeholders who took time out of their busy schedules to respond thoughtfully to this survey. We thank the SAR SIG-Edu members for their support of this endeavor.

## Author Contributions

**Conceptualization:** Heidi Most, Lisa Conboy, Rosaleen Ostrick, Belinda J. Anderson.

**Investigation:** Heidi Most, Lisa Conboy, Rosaleen Ostrick, Belinda J. Anderson.

**Methodology:** Heidi Most, Lisa Conboy, Rosaleen Ostrick, Belinda J. Anderson.

**Project administration:** Heidi Most.

**Supervision:** Heidi Most.

**Validation:** Heidi Most, Lisa Conboy, Rosaleen Ostrick, Belinda J. Anderson.

**Writing – original draft:** Heidi Most.

**Writing – review & editing:** Heidi Most, Lisa Conboy, Rosaleen Ostrick, Belinda J. Anderson.

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
