## [Decision Letter · Decision Letter 0]

6 Aug 2024

PONE-D-24-07771“Bringing greater research fluency into our educational vision”: A Qualitative Research Study on Improving East Asian Medicine Research EducationPLOS ONE

Dear Dr. Most,

Thank you for submitting your manuscript to PLOS ONE. After careful consideration, we feel that it has merit but does not fully meet PLOS ONE’s publication criteria as it currently stands. Therefore, we invite you to submit a revised version of the manuscript that addresses the points raised during the review process.

 The reviewers have provided comments for strengthening you work - specifically they have requested more clarity about context and consideration of the suitability of the EBM framework to traditional medical systems.

We look forward to receiving your revised manuscript.

Kind regards,

Jenny Wilkinson, PhD

Academic Editor

PLOS ONE

Reviewers' comments:

Reviewer's Responses to Questions

**Comments to the Author**

1. Is the manuscript technically sound, and do the data support the conclusions?

Reviewer #1: Partly

Reviewer #2: Yes

2. Has the statistical analysis been performed appropriately and rigorously? 

Reviewer #1: Yes

Reviewer #2: N/A

3. Have the authors made all data underlying the findings in their manuscript fully available?

Reviewer #1: Yes

Reviewer #2: No

4. Is the manuscript presented in an intelligible fashion and written in standard English?

Reviewer #1: Yes

Reviewer #2: Yes

5. Review Comments to the Author

Reviewer #1: The title should state clearly that it about Traditional Chinese Medicine (TCM) or Acupuncture. That would convey the readers what to expect from the article. Eastern Asian Medicine is a very broad term, hence the tile and the theme of the article doesn't match. Introduction could focus on what is TCM, research challenges, where all it is being practiced and what are the hurdles in research, followed by aim of the study. The material and method could be more logically explained and some details are more relevant to be added as a supporting additional material, rather than the main body of the methodology section.

While the themes identified as fair, it is impossible to include research and cover all of its aspects in any curriculum. That is the reason why people opt for higher courses. Hence a few expectations from the participants are better suited for additional skill appraisals, rather than in the course curriculum. Even though you have tried to build upon the EBM, please bear in mind, that as far as alternative medicines are concerned, they necessarily do not fit into the same framework as modern medicine does. For example, in TCM, many aspects are subjective and cannot be actually researched by the methods which are usually followed. These are called traditional, since they have been written in texts and being practiced for many years (even when scientific journals were not available).

Overall, the topic does not necessarily add anything new, which TCM experts are not aware about. The curriculum in China and adjacent nations have been much appreciated for the extensive coverage including a stress on research.

Reviewer #2: The topic area is very interesting and it would be helpful to traditional medicine education. Since this is a qualitative research, it would be important to understand the subjects involved in this study, that is, SMEs and stakeholders. It would be helpful to understand the context by providing a table summarizing the characteristics of the SMEs and stakeholders, for example, their age, gender, education level, professionals, their possible conflict of interest with the topic area. Since the curriculum addressed evidence-based medicine, so for EAM students, maybe evidence-based practice will be critically important for them to develop relevant skills such as structuring clinical questions with PICOs, and finding the literature, critical appraisal, and applying the evidence.

6. PLOS authors have the option to publish the peer review history of their article (what does this mean?). If published, this will include your full peer review and any attached files.

Reviewer #1: No

Reviewer #2: No

---

## [Author Response · Author response to Decision Letter 0]

25 Sep 2024

Please see the uploaded Response to Reviewers for a clear table.

Dear Editor and reviewers, 

Thank you very much for the opportunity to resubmit our revised manuscript. Your comments and suggestions were very helpful and have improved the quality of our work. Our responses to your suggestions are below. 

Comments/Suggestions Responses Text location

Editor 

The reviewers have provided comments for strengthening your work - specifically they have requested more clarity about context and consideration of the suitability of the EBM framework to traditional medical systems.

 Thank you for drawing our attention to these important points. We have provided additional context and clarity through an introductory first paragraph. We have addressed the issue of the suitability of the EBM framework to traditional medical systems by including evidence-based research that better captures the individualized and subjective nature of TCM. Please see our responses to specific suggestions below. Introduction lines 51-60 and S1

Reviewer 1 

1.The title should state clearly that is about TCM or Acupuncture. 

Thank you for your suggestion. We understand that EAM is a broad term and we have changed the title of the paper and the term EAM throughout the paper to TCM, as this term is more common internationally. Only direct quotations containing EAM remain. These changes can be found throughout the paper

2.Introduction should focus on what is TCM, research challenges, where it is being practiced and what are the hurdles in research, followed by the aim of the study. 

Thank you for this sugggestion. A paragraph has been added to the introduction that addresses TCM terminology, where it is being practiced, and the aim of the study. We address research challenges elsewhere in the paper and it is a topic in our model curriculum.

These changes can be found: Introduction lines 51-60. Research challenges are detailed in lines 103-125 and 375-395 and it is a topic in Table 4.

3.The material and method could be more logically explained and some details are more relevant to be added as a supporting additional material, rather than the main body of the methodology section.

We agree that the methodology needed to be better explained and organized. We have moved the first paragraph and first table in the Methodology section to S4 and replaced the first paragraph with one sentence and a brief timeline. We have also removed a paragraph detailing the timeline of sending requests to SMEs and stakeholders to S4. See Survey Instrument section, lines 156-179.

4. While the themes identified as fair, it is impossible to include research and cover all of its aspects in any curriculum. That is the reason why people opt for higher courses. Hence a few expectations from the participants are better suited for additional skill appraisals, rather than in the course curriculum. 

We agree that covering all research aspects in any curriculum is impossible. The value of our research is that it helped to identify the important topics to introduce to U.S. acupuncture students. Some U.S. TCM schools do not have specific research curricula, and this list of topics can serve as a guideline for them. In our paper, we suggest that programs can cover these topics wherever in their curriculum they deem most appropriate, including in higher-level courses or degrees. Pleease see Lines 526-532

4.Even though you have tried to build upon the EBM, please bear in mind, that as far as alternative medicines are concerned, they necessarily do not fit into the same framework as modern medicine does. For example, in TCM, many aspects are subjective and cannot be actually researched by the methods which are usually followed. These are called traditional, since they have been written in texts and being practiced for many years (even when scientific journals were not available).

We agree that modern medical research methods such as randomized controlled trials cannot capture the individualized nature or the breadth and depth of TCM. We have included this point as a topic under Worldview that students need to learn. Our curriculum includes research that uses designs that better capture the subjective and individualized nature of TCM, such as qualitative research, pragmatic clinical trials, and observational research. See Introduction, lines 103-113.

Table 4. Topic II. Also, see for example S1 reference:

Taylor-Swanson, L., Altschuler, D., Taromina, K., Anderson, B., Bensky, D., Cohen, M., ... & Conboy, L. (2022). SEAttle-based research of Chinese herbs for COVID-19 study: A whole health perspective on Chinese herbal medicine for symptoms that may be related to COVID-19. Global Advances in Health and Medicine, 11, 21649561211070483.

5.Overall, the topic does not necessarily add anything new, which TCM experts are not aware about. The curriculum in China and adjacent nations have been much appreciated for the extensive coverage including a stress on research.

We understand your comment about not adding new content. We set out to design a course that included the most relevant topics for students studying in the US. Some TCM schools in the US do not have extensive research curricula and this paper can serve as a guideline for them. 

Reviewer 2: 

1.The topic area is very interesting and it would be helpful to traditional medicine education. 

Thank you for your interest in and support for this topic. 

2.Since this is a qualitative research, it would be important to understand the subjects involved in this study, that is, SMEs and stakeholders. It would be helpful to understand the context by providing a table summarizing the characteristics of the SMEs and stakeholders, for example, their age, gender, education level, professionals, their possible conflict of interest with the topic area. 

We agree that more information on the SMEs and stakeholders would be useful. As you have suggested, we have provided that information in S3 - Characteristics of Subject Matter Experts and Stakeholders. S3 Characteristics of Subject Matter Experts and Stakeholders

3.Since the curriculum addressed evidence-based medicine, so for EAM students, maybe evidence-based practice will be critically important for them to develop relevant skills such as structuring clinical questions with PICOs, and finding the literature, critical appraisal, and applying the evidence.

We agree with you that EAM students need to know basic research and research literacy skills such as devising PICOs and finding and appraising research. The first topic in our list is the Importance of Research and Evidence-Based Medicine. We have added the subtopics of Introduction to Evidence-

Based Medicine and Introduction to Research Literacy to more explicitly address where this content appears in our curriculum. In addition, as we work to implement this curriculum, we plan to highlight the wealth of resources that are available online from highly reputable sources. For example, the Academic Collaborative for Integrative Health Council has created the Project to Enhance Research Literacy, which provides links to resources on teaching evidence-informed practice. The SAR SIG-Edu, from which this current research comes, has made available to its members content outlining basic research skills and basic research literacy curricula. Educating research-literate EAM students is our top priority. See Table 4, Topic I, line 499

Introduction, lines 80-92

Discussion, lines 530-536

4.The PLOS Data policy requires authors to make all data underlying the findings described in their manuscript fully available without restriction, with rare exception (please refer to the Data Availability Statement in the manuscript PDF file). 

We have deposited our data with Figshare, a freely available, open-access publishing platform. 

Thank you for your time and attention to this manuscript, and for allowing us to respond to your comments and improve it. 

Sincerely, 

HM, Corresponding Author

---

## [Editor Report · Decision Letter 1]

1 Oct 2024

“Bringing greater research fluency into our educational vision”: A qualitative research study on improving Traditional Chinese Medicine research education

PONE-D-24-07771R1

Dear Dr. Most,

We’re pleased to inform you that your manuscript has been judged scientifically suitable for publication and will be formally accepted for publication once it meets all outstanding technical requirements.

Kind regards,

Jenny Wilkinson, PhD

Academic Editor

PLOS ONE

Additional Editor Comments (optional):

Thank you for your comments on reviewer suggestions and manuscript revisions; these have satisfactorily addressed the issues raised by the reviewers.
---

## [Editor Report · Acceptance letter]

7 Oct 2024

PONE-D-24-07771R1 

PLOS ONE

Dear Dr. Most, 

I'm pleased to inform you that your manuscript has been deemed suitable for publication in PLOS ONE. Congratulations! Your manuscript is now being handed over to our production team.

Kind regards, 

on behalf of

Dr Jenny Wilkinson 

Academic Editor

PLOS ONE